# Anti-inflammatory effects of neutral lipids, glycolipids, phospholipids from *Halocynthia aurantium* tunic by suppressing the activation of NF-κB and MAPKs in LPS-stimulated RAW264.7 macrophages

A-yeong Jang[1]☯, Weerawan Rod-in[2,3]☯, Chaiwat Monmai[2,3], Gyoung Su Choi[1], Woo Jung Park(ID)[1,2,3]*

1 Department of Wellness-Bio Industry, Gangneung-Wonju National University, Gangneung, Gangwon, Korea, 2 Department of Marine Food Science and Technology, Gangneung-Wonju National University, Gangneung, Gangwon, Korea, 3 East Coast Life Sciences Institute, Gangneung-Wonju National University, Gangneung, Gangwon, Korea

☯ These authors contributed equally to this work.
* pwj0505@gwnu.ac.kr

## Abstract

*Halocynthia aurantium* is a marine organism that has been considered a promising source for bio-functional materials. Total lipids were extracted from *H. aurantium* tunic, and then they were separated into neutral lipids, glycolipids, and phospholipids. In the present study, fatty acid profiles of three lipids and their anti-inflammatory effects in RAW264.7 cells were investigated. Among the lipid classes, phospholipids showed the diversity of fatty acid constituents, compared with the glycolipids and neutral lipids. Three lipids contain different contents of fatty acids depending on the kinds of lipids. The most contents were saturated fatty acids (SFAs, 53–69% of the fatty acids) and monounsaturated fatty acids (MUFAs, 15–17% of fatty acids) and polyunsaturated fatty acids (PUFAs, 14–32% of fatty acids) are followed. *H. aurantium* lipids not only dose-dependently inhibited nitric oxide production but also reduced the expression of inflammatory cytokine genes such as *TNF-α*, *IL-1β*, and *IL-6* in LPS-stimulated macrophages. It was also demonstrated that the expression of *COX-2* was dose-dependently suppressed. Moreover, *H. aurantium* lipids decreased phosphorylation of NF-κB p-65, p38, ERK1/2, and JNK, suggesting that three lipids from *H. aurantium* tunic provide anti-inflammatory effects through NF-κB and MAPK signaling. These results indicate that *H. aurantium* is a potential source for anti-inflammation.

## Introduction

Lipids, especially essential fatty acids play an important role in the health and development of humans and play a critical role in the prevention of disease by altering their composition [1]. Lipids are a source of energy and the structure of cell membranes and polyunsaturated fatty acids such as eicosapentaenoic acid (EPA) and docosahexaenoic acid (DHA) are essential fatty

**Data Availability Statement:** All relevant data are within the paper and its Supporting information files.

**Funding:** This study was partially supported by the Basic Science Research Program of the National Research Foundation of Korea (NRF), which is funded by the Ministry of Science, ICT & Future Planning (2019R1A2B5B01070542). This research project is also supported by the University Emphasis Research Institute Support Program (No.2018R1A61A03023584), which is funded by the National Research Foundation of Korea. The funders had no role in study design, data collection and analysis, decision to publish, or preparation of the manuscript.

**Competing interests:** The authors declare no conflict of interest.

acids to regulate the inflammatory responses on the macrophage cells, which are considered anti-inflammatory agents [2–5]. Moreover, they can inhibit the activation of the pro-inflammatory transcription factor, such as nuclear factor κB (NF-κB) [3, 5, 6] and suppress the activation of mitogen-activated protein kinases (MAPKs) signaling pathway [6, 7]. These pathways involved inflammatory responses, which produce inflammatory cytokines and inflammatory mediators. [8]. In addition, high levels of EPA and DHA were reported from the lipid extracts including from total lipids, neutral lipids, and polar lipids of various ascidian species [9–12]. The lipid extracts from ascidians have improved beneficial health effects, such as anti-diabetic [13] and antioxidant effects.

Macrophages play important roles in inflammation and they can be activated by endotoxin which cause the production of inflammatory cytokines such as interleukin-1β (IL-1β), IL-6, and tumor necrosis factor (TNF)-α, and releasing cyclooxygenase-2 (COX-2) and nitric oxide synthase (iNOS) catalyze the production of prostaglandin E2 (PGE$_2$) and nitric oxide (NO) as inflammatory mediators [8, 14]. The natural substances such as alkaloids, steroids, polysaccharides, fatty acids, proteins, and other, which were isolated form marine biomaterials have been demonstrated strong anti-inflammatory activities [15]. They were determined by inhibiting the production of NO, TNF-α and suppressing the expression of *IL-1β*, *IL-6*, *TNF-α*, *iNOS* and *COX-2*, in RAW264.7 macrophages activated by lipopolysaccharides (LPS) [16–19].

*Halocynthia aurantium* is a solitary ascidian found in the Southern, Eastern Sea of Korea, and Northern Sea of Japan. The tunic is an essential structure part in the outer protective covering of the body, which contains a cellulose-like substance [20, 21]. Total lipids of *H. aurantium* tunic composed of the most abundant of palmitic acids (16:0), stearic acids (18:0), α-linolenic acids (ALA, 18:3 n-3), eicosapentaenoic acids (EPA, 20:5 n-3), and docosahexaenoic acids (DHA, 22:6 n-3) [22]. Similar to ascidian species, *H. roretzi* showed also the same fatty acids in total lipids, neutral lipids, and phospholipids. *H. aurantium* showed biological effects such as antimicrobial peptides (Dicynthaurin, and Halocidin) [23–25], and antioxidant effects [26]. Vanadium-binding protein from *H. roretzi* was investigated in macrophage-like RAW264.7 cells stimulated by LPS [27, 28]. However, no previous study has evaluated the fatty acid profiles of individual lipids such as phospholipids, glycolipids, and neutral lipids, which were isolated from *H. aurantium* tunic. Moreover, few studies have determined how these fractionated lipids exert anti-inflammatory activity on macrophages.

Therefore, the present study was to identify the fatty acid composition in *H. aurantium* lipids, containing neutral lipids, glycolipids, and phospholipids, and their anti-inflammatory activities using RAW264.7 cells.

## Materials and methods

### Preparation of fractionated lipids from *H. aurantium* tunic

*H. aurantium* used in this study was obtained in Jumunjin market on the he East Sea near Gangwon Province, South Korea. *H. aurantium* tunic was dried and homogenized to powder. Total lipids were extracted using a modified method by Bligh and Dyer [29]. A mixture of chloroform/methanol (1:2, v/v) containing 0.01% of butylated hydroxytoluene (BHT) to the solvent as antioxidant [30] was added to 4.5 g of dry weight sample and centrifuged at 3000 rpm for 10 minutes. Subsequently, the organic solvent was collected and filtered. A rotary evaporator (IKA® RV10, EYELA, China) was to remove the solvent, and the residual solvent was removed by nitrogen evaporator (N-EVAP, Organomation Associates Inc., USA). The extracted lipids with a yield of 33.6 mg (w/w) or 0.75% of dry material were then resuspended in hexane for fractionation of lipid extracts.

The total lipids were added into silica gel column filled with silica gel and anhydrous sodium sulfate using chloroform, acetone, and methyl alcohol to produce the neutral lipids, glycolipids, and phospholipids, respectively. After separating, these extracts were evaporated and weighed. The neutral lipids, glycolipids, and phospholipids were shown to have a high lipid yield of 8.63% (2.9 mg), 30.95% (10.4 mg), and 44.94% (15.1 mg) of total lipid, respectively. The *H. aurantium* lipids at 2 mg/mL (set to 100%) were prepared in dimethyl sulfoxide (DMSO, Sigma-Aldrich, USA, Cat# D8418) and stored at −20 ◦C for further analysis.

## Analysis of fatty acid profiles

Fatty acid compositions of neutral lipids, glycolipids, and phospholipids were determined using gas chromatography (GC)-flame ionization detection (FID) (Perkin Elmer, Waltham, MA, USA) as previously described [31]. Quantification of fatty acid peaks were identified by the comparison of their retention times with heptadecanoic acid (C17:0) as internal standard (Sigma-Aldrich, USA, Cat# H3500). Results were presented as quintuplate (*n* = 5) independent experiments.

## Cell culture and treatment

Mouse macrophages RAW 264.7 cell line was obtained from Koran Cell Line Bank (KCLB, Cat# 40071, RRID: CVCL_0493). The cells were culture in RPMI-1640 medium (Gibco™, Waltham, USA, Cat# 11875–093) supplemented with 10% fetal bovine serum (FBS, Welgene, Korea, Cat# S001-07) and 1% penicillin/streptomycin (Welgene, Korea, Cat# LS202-02), and then incubated at 37°C in a humidified atmosphere of 5% $CO_2$. The lipids were dissolved in RPMI-1640 medium (GibcoTM, Waltham, USA, Cat# 11835–030) supplemented with 1% FBS and 1% penicillin/streptomycin to different concentrations at 0.5%, 1.0%, 2.0% and 4.0%. 100 μL of the lipids were culture into the RAW264.7 cells (at a density of $1 \times 10^5$ cell/well) for 1 h. After incubation, the presence or absence of 1 μg/mL lipopolysaccharide (LPS from *Escherichia coli* O111:B4, Sigma-Aldrich, USA, Cat# L4391-1MG) were added into each well for another 24 h.

## Measurement of cell viability and NO production

The cell viability of three lipids from *H. aurantium* was analyzed using EZ-Cytox Cell Viability Assay Kit (DaeilLab Service, Seoul, Korea, Cat# EZ-3000) as described by Kim et al. [19]. Three independent experiments were performed in triplicate. Griess reagent (Promega, WI, USA, Cat# G2930) was used to evaluate the LPS induced the production of nitric oxide [32] according to the manufacturer's instructions. Three independent experiments were performed in triplicate.

## Analysis of immune gene expression by quantitative real-time PCR

The mRNA expression levels of immune-regulated genes were determined by qRT-PCR. TRI reagent® (Molecular Research Center, Cincinnati, OH, USA, Cat# TR118) was used to extract the total RNA from RAW264.7cells. High capacity cDNA reverse transcription kit (Applied Biosystems, Foster City, CA, USA, Cat# 4368814) was used to reverse transcribe cDNA. Real-time PCR was performed on QuantStudio™ 7 FlexReal-Time PCR System (Applied Biosystems, Foster City, CA, USA) using TB Green® Premix Ex Taq™ II (Takara Bio Inc., Shiga, Japan, Cat#RR820A). The relative expression levels of *IL-1β*, *IL-6*, *TNF-α*, and *COX-2* were normalized using the *β-actin* (S1 Table). Results were presented as triplicate independent experiments.

## Western blotting analysis

Cell lysates were prepared using RIPA buffer (Tech & Innovation, Hebei, China, Cat# BRI-9001) containing 0.5 mM EDTA solution, and a protease & phosphatase inhibitor cocktail (Thermo Fisher Scientific, USA, Cat# 78440). SDS-PAGE and western blotting were performed. The protein was analyzed by immunoblot using primary antibodies against phospho-nuclear NF-κB-p65 (Cell Signaling Technology, MA, USA, Cat# 3033, RRID: AB_331284), phospho-p38 (Cell Signaling Technology, MA, USA, Cat# 9211, RRID:AB_331641), phospho-ERK1/2 (Cell Signaling Technology, MA, USA, Cat# 9101, RRID: AB_331646), phospho-JNK (Cell Signaling Technology, MA, USA, Cat# 9251, RRID: AB_331659), and α-tubulin (Abcam, Cambridge, UK, Cat# ab15246, RRID:AB_301787), and then was incubated with secondary antibodies as goat anti-rabbit IgG (H+L)-HRP (GenDEPOT, TX, USA, Cat# SA006-500). The protein bands were measured by the ChemiDoc XRS+ imaging system, and ImageLab software (Bio-Rad, Hercules, CA, USA). Results were presented as triplicate independent experiments.

## Statistical analysis

All data were subjected to analysis of variance using Statistix 8.1 Statistics Software (Tallahassee, FL, USA). One-way ANOVA followed by the Duncan's multiple range test was used to evaluate the significance of the differences ($p < 0.05$). Data are expressed as mean ± standard deviation (SD).

# Results

## Fatty acid profiles of neutral lipids, glycolipids, and phospholipids, which were isolated from *H. aurantium* tunic

Fig 1 presents the percentages of fatty acids composition in neutral lipids, glycolipids, and phospholipids, which determined by GC-FID analyses. The results showed that the highest amount of the lipids were SFAs in neutral lipids (63.75%), glycolipids (52.70%), and phospholipids (69.45%). Total amount of MUFAs were 16.92% in neutral lipids, 14.87% in glycolipids, and 16.52% phospholipids. Moreover, total amount of PUFAs are 19.34, 32.44, 14.03% of neutral lipids, glycolipids, and phospholipids, respectively.

At first, myristic acid (14:0), palmitic acid (16:0), and stearic acid (18:0) was mainly contained in SFAs, and in addition the phospholipids also showed arachidic acid (20:0). At second,

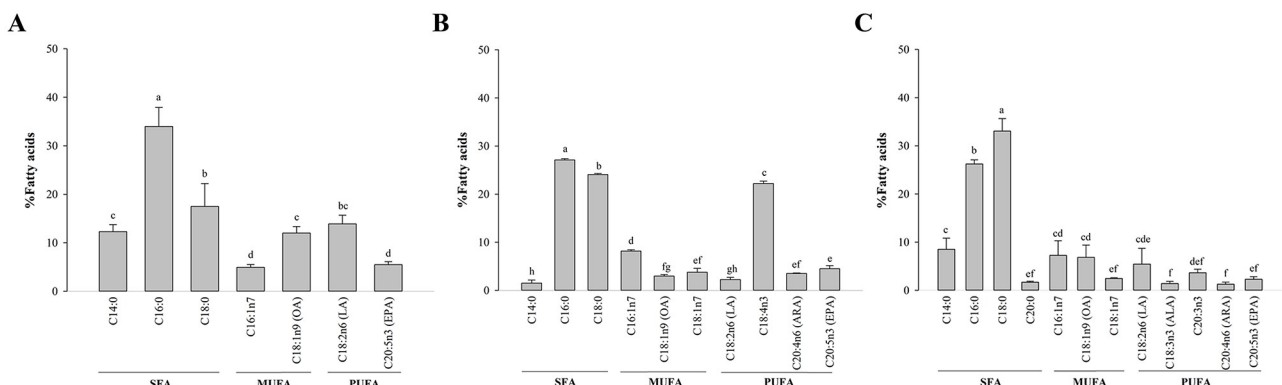

**Fig 1. Fatty acid composition (%) of *H. aurantium* tunic lipids.** (A) Neutral lipids. (B) Glycolipids. (C) Phospholipids. The letters a–h indicate significant differences ($p < 0.05$) between the amounts of fatty acids, which were obtained from each lipid of *H. aurantium* tunic. Results represent means ± SD (*n* = 5). SFA, saturated fatty acid; MUFA, monounsaturated fatty acid; PUFA, polyunsaturated fatty acid.

the major MUFAs are palmitoleic acid (16:1n-7) and oleic acid (18:1n-9) and they were contained with 4.92 and 11.99% in neutral lipids, 8.16 and 2.96% in glycolipids, and 7.25 and 6.82% in phospholipids, respectively. At third, the major PUFAs are linoleic acid (18:2n-6) and eicosapentaenoic acid (20:5n-3) which were 13.87 and 5.46% in neutral lipids, 2.24 and 4.49% in glycolipids, and 5.43 and 2.29% in phospholipids. Especially, the presence of stearidonic acid (18:4n-3) and arachidonic acid (20:4n-6) was also identified in glycolipids (22.20 and 3.51%) and phospholipids (3.64 and 1.27%).

## Effect of neutral lipids, glycolipids, and phospholipids from *H. aurantium* tunic on macrophage cell proliferation

To determine if the fractionated lipids containing neutral lipids, glycolipids, and phospholipids are not toxic to RAW264.7 macrophages, we investigated cytotoxicity using an EZ-Cytox cell viability assay kit. As shown in Fig 2A, our results showed that the neutral lipids, glycolipids, and phospholipids did not give any cytotoxicity at concentrations of 0.5–4.0% of lipids.

## Anti-inflammatory effects of neutral lipids, glycolipids, and phospholipids from *H. aurantium* tunic on NO production

To investigate the anti-inflammatory activity of fractionated lipids from *H. aurantium* tunic, we measured lipid-mediated inhibition of NO production in LPS-stimulated RAW264.7 cells using Griess reagent assay. NO production was determined using different of three lipids at 0.5–4.0% concentrations. Neutral lipids, glycolipids, and phospholipids of *H. aurantium* tunic gradually decreased NO production in LPS-stimulated RAW264.7 cells according to the lipid concentration (Fig 2B).

## Anti-inflammatory effects of neutral lipids, glycolipids and phospholipids from *H. aurantium* tunic on immune-associated gene expression

Since NO production was significantly inhibited by the fractionated lipids from *H. aurantium* tunic, we measured the mRNA expression of immune-associated genes such as *IL-1β*, *IL-6*, *TNF-α*, and *COX-2* in LPS-stimulated RAW264.7 cells by qRT-PCR. The expression levels of inflammatory cytokines were significantly down-regulated depending on the concentration of *H. aurantium* lipids, including neutral lipids (Fig 3A), glycolipids (Fig 3B), and phospholipids

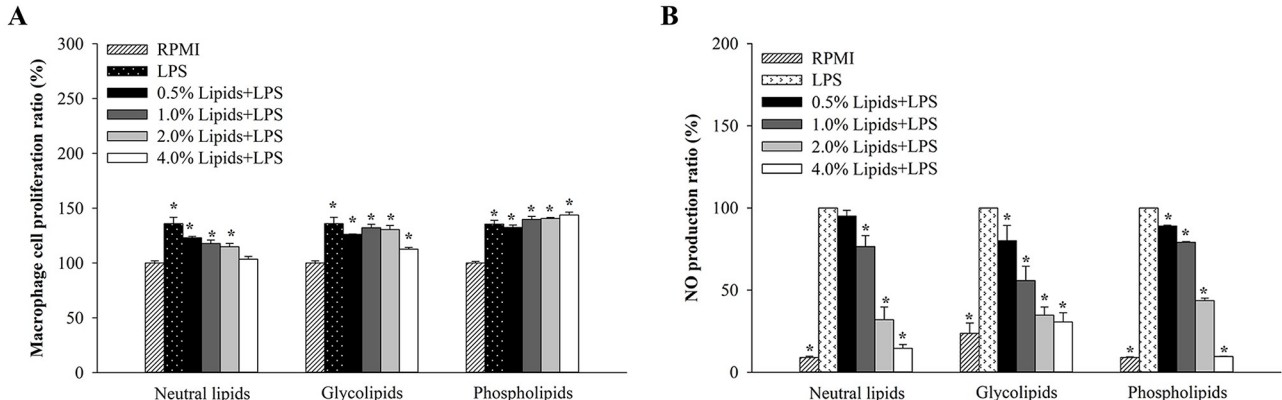

**Fig 2. The effects of neutral lipids, glycolipids and phospholipids from *H. aurantium* tunic on macrophage proliferation.** (A) Cell proliferation. (B) NO production. Significant different at *p*<0.05 (*) compared with RPMI.

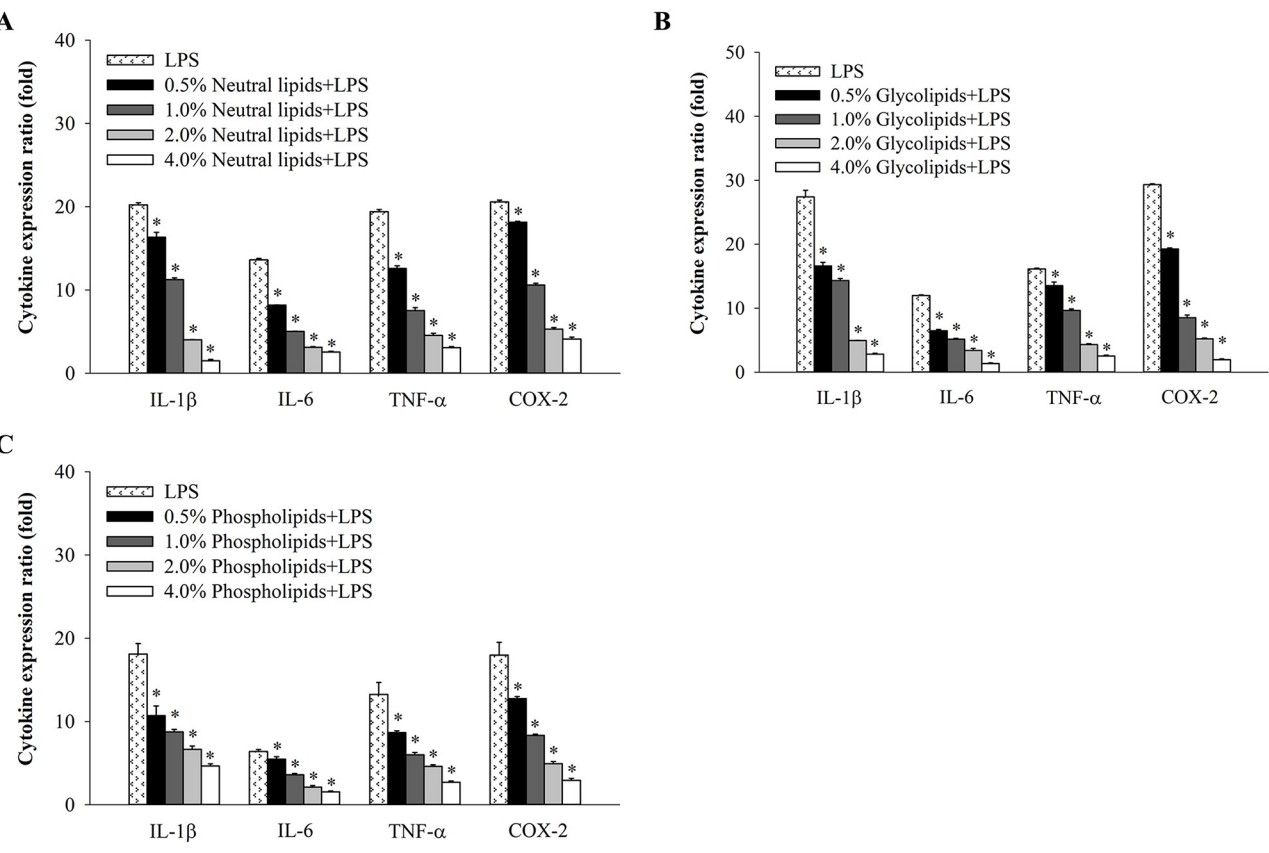

**Fig 3. Quantification of immune-associated gene expression (fold) in LPS-stimulated RAW264.7 cells.** (A) The relative mRNA expression of neutral lipids. (B) The relative mRNA expression of glycolipids. (C) The relative mRNA expression of phospholipids. Significant different at $p < 0.05$ (*) compared with LPS.

(Fig 3C). Our results showed that the *IL-1β* expression was highly reduced by *H. aurantium* lipids and the expression levels of other cytokine genes such as *IL-6* and *TNF-α* were dose-dependently decreased. Moreover, the expression levels of *COX-2*, another well-known inflammatory biomarker, were dose-dependently suppressed according to the concentration of neutral lipids, glycolipids, and phospholipids from *H. aurantium* tunic.

## Anti-inflammatory effects of neutral lipids, glycolipids and phospholipids from *H. aurantium* tunic on MAPK and NF-κB signaling pathway

In order to investigate the understanding of the molecular mechanism by which *H. aurantium* lipids exert their anti-inflammatory effect, immune signaling pathways such as NF-κB and MAPK were analyzed. Our results indicated neutral lipids (Fig 4A), glycolipids (Fig 4B), and phospholipids (Fig 4C), which were isolated from *H. aurantium* tunic, dose-dependently inhibited the phosphorylation of NF-κB p-65, ERK1/2, JNK, and p38 in a dose-dependent manner. These results showed that fractionated lipids from *H. aurantium* tunic inhibited inflammation through MAPK and NF-κB signaling pathways in LPS-stimulated RAW246.7 cells.

## Discussion

*Halocynthia aurantium*, an edible ascidian species, has not been studied, although tunicates and ascidians are well-known to contain biologically active compounds. This study was

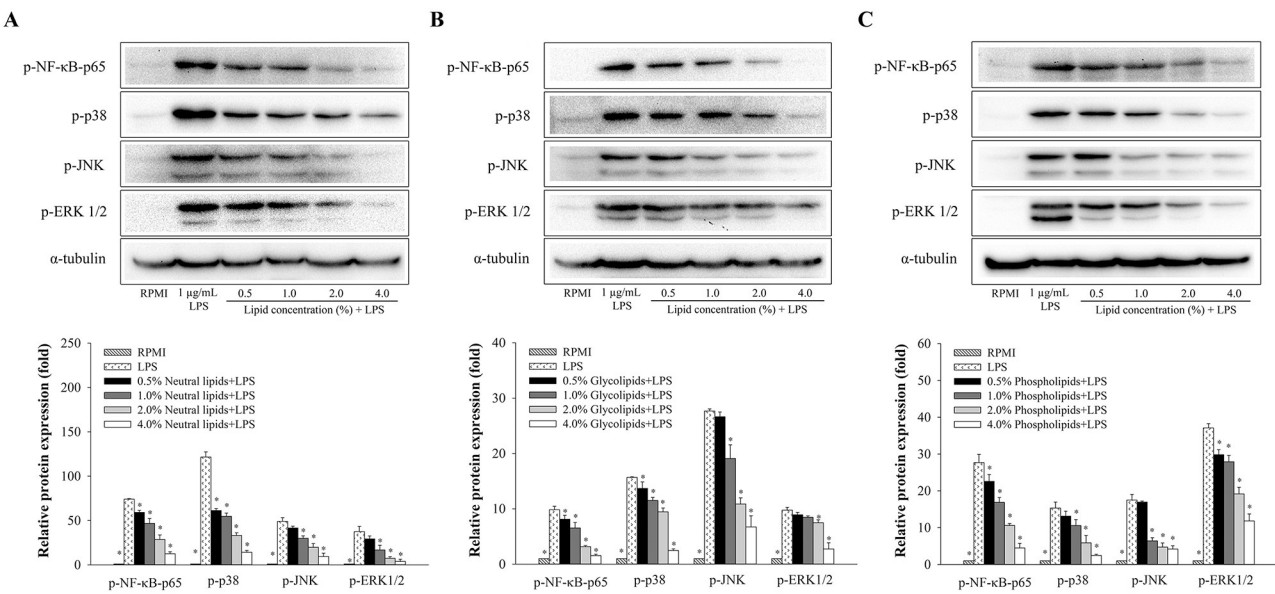

**Fig 4. The effects of fractionated lipids from *H. aurantium* tunic on the protein expression associated with NF-κB and MAPK pathways in LPS-stimulated RAW264.7 cells.** (A) Western blot and relative band of neutral lipids. (B) Western blot and relative band blot of glycolipids. (C) Western blot and relative band of phospholipids. Significant different at $p < 0.05$ (*) compared with LPS.

undertaken to analyze the fatty acid composition in fractionated lipids including neutral lipids, glycolipids, and phospholipids from *H. aurantium* tunic, and to investigate their anti-inflammatory effects on LPS-stimulated macrophages.

Recently, the total lipids of *H. aurantium* tunic were analyzed the fatty acid compositions, consisting of the most abundant of palmitic acids (21.73±2.16), stearic acids (33.13±3.22), oleic acid (6.78±0.28%), LA (2.72±0.23%), dihomo α-linolenic acid (4.09±0.36), EPA (3.88 ±0.31%), and DHA (3.38±0.34%) [31]. The current results showed the fatty acid profiles of fractionated lipids (neutral lipids, glycolipids, and phospholipids) from the total lipids extracted from *H. aurantium* tunic (Fig 1).

The lipid extracts including neutral lipids, glycolipids, and phospholipids, which isolated from *H. roretzi* were found to be similar to the total lipids of *H. aurantium* [12]. In addition, the lipid extracts of *Hippocampus trimaculatus*, containing neutral lipids, glycolipids, phospholipids reduced the production of NO, IL-6, IL-1β, and TNF-α in LPS-stimulated RAW264.7 cells [18]. Moreover, many studies investigated the anti-inflammatory activities of various bioactive compounds from ascidian species. The enzymatic hydrolysates derived from *Styela clava* down-regulated LPS-induced expression of *iNOS* and *COX-2*, suppressed the production of pro-inflammatory cytokines, including IL-1β, IL-6, and TNF-α and they inhibited LPS-induced phosphorylation of ERK, JNK, and p38 [33]. Thomson et al. reported that polysaccharides isolated from the *Ascidiella aspersa* exhibited anti-inflammatory effects in *vitro* and in *vivo* systems [34]. *H. roretzi* also exhibited the anti-inflammatory effect of carotenoids and vanadium-binding protein, which down-regulated the expression of pro-inflammatory cytokines, including *IL-1β*, *IL-6*, and *TNF-α* as well as *iNOS* and *COX-2* mRNA expression on LPS-stimulated RAW264.7 cells [28]. Similar to previous reports, our results also showed that *H. aurantium* lipids including neutral lipids, glycolipids, and phospholipids effectively suppressed the expression of inflammatory cytokines, such as *IL-1β*, *IL-6*, and *TNF-α* (Fig 3) which are pro-inflammatory cytokines that activated T cells, maturation of B cells, and

activation of NK cells, especially monocytes or macrophages [35]. In addition, the COX-2 expression levels, which is a key enzyme in the production of prostaglandins (PGE$_2$) by LPS-stimulated macrophages in the inflammatory process [36], was down-regulated by *H. aurantium* lipids depending on the lipid concentration (Fig 3). These results suggested that *H. aurantium* lipids are biomaterials to contain anti-inflammatory effects on immune systems in a physiological system.

In immune signaling pathways, there are critical two pathways including NF-κB and MAPK pathways, in which NF-κB is a transcription factor, is a critical regulator mediator for iNOS, COX-2 transcription, and the production of cytokines in LPS-induced macrophages [37, 38], and The MAPK pathway is considered one of the main intracellular signaling pathways that regulate inflammatory responses [39]. Our current results showed that three lipids from *H. aurantium* tunic reduced the activation of the NF-κB pathway by inhibiting the phosphorylation of the NF-κB p-65 subunit (Fig 4). In addition, *H. aurantium* lipids inhibited the expression of MAPKs (ERK1/2, JNK, and p38) in LPS-induced RAW 264.7 cells in a dose-dependent manner. Similar to ascidian species, vanadium-binding protein from *H. roretzi* has reported being anti-inflammatory effects. This protein inhibited the LPS-stimulated inflammatory response in RAW264.7 macrophages through NF-κB and MAPK pathways [27]. Skipjack tuna eyeball oil was identified as the main fatty acids of DHA (25%) and EPA (5%), inhibited the production of NO and pro-inflammatory cytokine by suppressing the activation of NF-κB and MAPK signaling pathways in RAW264.7 cells [40]. Taken together, these results indicated that three lipids of *H. aurantium* tunic induced the suppression of NO, and immune-regulated genes in activated macrophages through the inhibition of NF-κB and MAPK pathways.

## Conclusions

The present study demonstrated that the fractionated lipids from *H. aurantium* tunic, including neutral lipids, glycolipids, and phospholipids composed of the highest contents of 16:0 and 18:0 as SFAs, 16: 1n7 and 18:1n9 as MUFAs, and 18:2n6 and 20:5n3 as PUFAs. Three lipids of *H. aurantium* tunic significantly inhibited the production of NO and the expression of immune-associated genes such as *IL-1β*, *IL-6*, *TNF-α*, and *COX-2*. Likewise, the decreased expression levels led to further activation of NF-κB p-65 and MAPK molecules, such as ERK1/2, JNK, and p38, thus alleviating the immune response. These results might be helpful to understand the anti-inflammatory mechanisms of *H. aurantium* lipids on immune cells and suggested that *H. aurantium* is a potential source for anti-inflammation.

## Supporting information

**S1 Raw images. Original western blot gel image data.**
(PDF)

**S1 Table. Nucleotide primers used in this study.**
(PDF)

## Author Contributions

**Conceptualization:** Chaiwat Monmai, Woo Jung Park.

**Data curation:** Weerawan Rod-in, Chaiwat Monmai, Woo Jung Park.

**Formal analysis:** A-yeong Jang, Weerawan Rod-in, Chaiwat Monmai.

**Funding acquisition:** Woo Jung Park.

**Investigation:** A-yeong Jang.

**Methodology:** A-yeong Jang, Weerawan Rod-in, Gyoung Su Choi.

**Project administration:** Woo Jung Park.

**Resources:** Woo Jung Park.

**Software:** A-yeong Jang, Weerawan Rod-in.

**Supervision:** Woo Jung Park.

**Validation:** A-yeong Jang, Weerawan Rod-in, Chaiwat Monmai.

**Visualization:** A-yeong Jang, Weerawan Rod-in.

**Writing – original draft:** A-yeong Jang.

**Writing – review & editing:** Weerawan Rod-in, Chaiwat Monmai, Woo Jung Park.

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
