## [Decision Letter · Decision Letter 0]

20 Apr 2022

PONE-D-22-03751Anti-inflammatory effects of neutral lipids, glycolipids, phospholipids from Halocynthia aurantium tunic by suppressing the activation of NF-κB and MAPKs in LPS-stimulated RAW264.7 macrophagesPLOS ONE

Dear Dr. Park,

Thank you for submitting your manuscript to PLOS ONE. After careful consideration, we feel that it has merit but does not fully meet PLOS ONE’s publication criteria as it currently stands. Therefore, we invite you to submit a revised version of the manuscript that addresses the points raised during the review process.

Please revise your manuscript taking into account the comments of the two reviewers. Point 2 about figure quality from the first reviewer can be disregarded as they did not download the full tiff files.

We look forward to receiving your revised manuscript.

Kind regards,

C. Michael Greenlief, Ph.D.

Academic Editor

PLOS ONE

Journal Requirements:

2. In your Methods section, please provide additional details regarding the H. aurantium used in your study and ensure you have described the source. For more information regarding PLOS' policy on materials sharing and reporting, see https://journals.plos.org/plosone/s/materials-and-software-sharing#loc-sharing-materials.

(This study was partially supported by the Basic Science Research Program of the National Research Foundation of Korea (NRF), which is funded by the Ministry of Science, ICT & Future Planning (2019R1A2B5B01070542). This research project is also supported by the University Emphasis Research Institute Support Program (No.2018R1A61A03023584), which is funded by the National Research Foundation of Korea.)

Additional Editor Comments:

Please revise your manuscript taking into account the comments of the two reviewers. Point 2 about figure quality from the first reviewer can be disregarded as they did not download the full tiff files.

Reviewers' comments:

Reviewer's Responses to Questions

**Comments to the Author**

1. Is the manuscript technically sound, and do the data support the conclusions?

Reviewer #1: Partly

Reviewer #2: Partly

2. Has the statistical analysis been performed appropriately and rigorously? 

Reviewer #1: Yes

Reviewer #2: Yes

3. Have the authors made all data underlying the findings in their manuscript fully available?

Reviewer #1: Yes

Reviewer #2: Yes

4. Is the manuscript presented in an intelligible fashion and written in standard English?

Reviewer #1: Yes

Reviewer #2: Yes

5. Review Comments to the Author

Reviewer #1: The manuscript describes anti-inflammatory effects of lipid extracted from Halocynthia aurantium. Even though results are interesting, need major revision to enhance the quality of the paper sutiable for publication in PLOS ONE.

Major comments –

1. Author described the extraction method of total lipids (lines 86-87), unfortunately no extraction yield was given anywhere. Similarly, lipid extracted from H. aurantium was fractionated into neutral lipid, glycolipid and phospholipids (line 94-96), and neither initial lipid weight nor the fractions yield were given. This is important information to understand the value of the lipid from H. aurantium.

2. Figures are in poor quality possible due to electronic submission process.

3. Figure 1 described the percentage of individual fatty acids in terms of total fatty acid count. It will not give exact percentage of the fatty acid in the tested fractions i.e., neutral lipid, glycolipids or phospholipids. Lipid fraction such as neutral lipids may contain other components too, better to present the fatty acid data in w/w basis.

4. No positive control was used in any of the bio-assays. Recommended to have positive control for comparison.

5. Author described the final concentration of the lipids 20 mg/ml (line 97), but in Figures 2-4 results are described in percentage lipids, need further clarification how the test sample were prepared from 20 mg/ml to 0.5-4% lipids.

6. At 4.0% lipid concentration neutral lipid and glycolipids fractions showed cytotoxicity effect (Figure 2A, lines 171-174). No explanation was given why these lipids fractions were tested at 4.0% for NO and other assays. NO inhibition and other positive effect at this concentration may due to cell death.

Reviewer #2: This manuscript describes a series of studies looking at the anti-inflammatory effects of lipids extracted form Halocynthia aurantium. Overall the results are interesting, but revisions are needed to help improve the manuscript. These points are listed below.

1. The authors describe an extraction method in lines 86-98. This methodology needs to be more quantitative. How much material was used to perform the extract? What was the mass obtained in each fraction? This will allow one to know the lipid content in the extraction. While each fraction analyzed had the same mass of lipids, it would be more useful to have the number on a weight percent basis.

2. There needs to be a positive control in the bioassays.

3. Why was a 4.0% lipid concentration chosen for the results in Figure 2? No rationale for this choice is given and should be provided.

6. PLOS authors have the option to publish the peer review history of their article (what does this mean?). If published, this will include your full peer review and any attached files.

Reviewer #1: No

Reviewer #2: No

---

## [Author Response · Author response to Decision Letter 0]

2 Jun 2022

Reviewer #1

The manuscript describes anti-inflammatory effects of lipid extracted from Halocynthia aurantium. Even though results are interesting, need major revision to enhance the quality of the paper suitable for publication in PLOS ONE.

Major comments 

1. Author described the extraction method of total lipids (lines 86-87), unfortunately no extraction yield was given anywhere. Similarly, lipid extracted from H. aurantium was fractionated into neutral lipid, glycolipid and phospholipids (line 94-96), and neither initial lipid weight nor the fractions yield was given. This is important information to understand the value of the lipid from H. aurantium.

- It was corrected in Line 87-88 and 92-95.

- Total lipids produced a yield of 33.6 mg (w/w) or 0.75% of dry material (4.5 g). If 4.5 g of dried material is 100%, yield (33.6 mg) is 0.75% of dried material (4.5 g).

- The neutral lipids, glycolipids, and phospholipids were shown to have a high lipid yield of 8.63 (2.9 mg), 30.95 (10.4 mg), and 44.94 (15.1 mg) % of total lipid (33.6 mg), respectively.

- Thanks.

2. Figures are in poor quality possible due to electronic submission process.

- It was corrected.

- Thanks.

3. Figure 1 described the percentage of individual fatty acids in terms of total fatty acid count. It will not give exact percentage of the fatty acid in the tested fractions i.e., neutral lipid, glycolipids or phospholipids. Lipid fraction such as neutral lipids may contain other components too, better to present the fatty acid data in w/w basis.

- Based on an internal standard, fatty acids were analyzed.

- We are attaching the percentage of fatty acids like the following.

- We calculated all detected lipids by the equation as (each lipid/all fatty acids) * 100.

- Thanks.

4. No positive control was used in any of the bio-assays. Recommended to have positive control for comparison.

- Normally the anti-inflammatory effects used LPS as positive control.

- Thanks. 

- References

o Fernando, I. S., Sanjeewa, K. A., Kim, H. S., Kim, S. Y., Lee, S. H., Lee, W. W., & Jeon, Y. J. (2017). Identification of sterols from the soft coral Dendronephthya gigantea and their anti-inflammatory potential. Environmental Toxicology and Pharmacology, 55, 37-43.

o Li, M., Zhang, L., Cai, R. L., Gao, Y., & Qi, Y. (2012). Lipid‐soluble extracts from Salvia miltiorrhiza inhibit production of LPS‐induced inflammatory mediators via NF‐κB modulation in RAW264.7 cells and perform anti-inflammatory effects in vivo. Phytotherapy Research, 26(8), 1195-1204.

o Kim, K. N., Ko, Y. J., Yang, H. M., Ham, Y. M., Roh, S. W., Jeon, Y. J., ... & Oda, T. (2013). Anti-inflammatory effect of essential oil and its constituents from fingered citron (Citrus medica L. var. sarcodactylis) through blocking JNK, ERK and NF-κB signaling pathways in LPS-activated RAW 264.7 cells. Food and Chemical Toxicology, 57, 126-131.

o Wang, T., Wu, F., Jin, Z., Zhai, Z., Wang, Y., Tu, B., ... & Tang, T. (2014). Plumbagin inhibits LPS-induced inflammation through the inactivation of the nuclear factor-kappa B and mitogen activated protein kinase signaling pathways in RAW 264.7 cells. Food and chemical toxicology, 64, 177-183.

5. Author described the final concentration of the lipids 20 mg/ml (line 97), but in Figures 2-4 results are described in percentage lipids, need further clarification how the test sample were prepared from 20 mg/ml to 0.5-4% lipids.

- It was corrected in Line 99.

- The stock of lipids has some mistakes in that the final concentration of lipids is 2 mg/mL.

- Stock of lipids has concentration at 2 mg/mL (w/v) as 100%, and then lipids were diluted to 0.5, 1.0, 1.5 and 2.0% (v/v) with medium. 

- Thanks.

6. At 4.0% lipid concentration neutral lipid and glycolipids fractions showed cytotoxicity effect (Figure 2A, lines 171-174). No explanation was given why these lipids fractions were tested at 4.0% for NO and other assays. NO inhibition and other positive effect at this concentration may due to cell death.

- Figure 2A showed cell proliferation in RAW264.7 cells (in the absence of LPS) that are toxic at high concentrations while cell proliferation in LPS-stimulated RAW264.7 cells showed non-toxicity.

- The purpose of our study is to test the anti-inflammatory effects. Therefore, we tested the three lipids with the addition of LPS. It showed the highest NO production at a concentration of 4%, especially phospholipids, which reduced NO production by 9.52 % while causing no toxic in the cells. 

- We changed the previous one to new figure 2A including the addition of LPS.

- In our study, the three lipids showed the highest NO production at a concentration of 4%, especially phospholipids, which reduced NO production by 9.52 % while causing no toxic in the cells.

- All experiments with three lipids should use the same concentration. 

- Following references showed the cell viability of a sample to be lower than control (≤ 80%), and that they can also be evaluated for NO production, gene expression, and protein expression.

- References

o Qiao, J., Xu, L. H., He, J., Ouyang, D. Y., & He, X. H. (2013). Cucurbitacin E exhibits anti-inflammatory effect in RAW 264.7 cells via suppression of NF-κB nuclear translocation. Inflammation Research, 62(5), 461-469.

o Konishi, I., Hosokawa, M., Sashima, T., Maoka, T., & Miyashita, K. (2008). Suppressive effects of alloxanthin and diatoxanthin from Halocynthia roretzi on LPS-induced expression of pro-inflammatory genes in RAW264. 7 cells. Journal of Oleo Science, 57(3), 181-189.

o Orecchini, E., Mondanelli, G., Orabona, C., Volpi, C., Adorisio, S., Calvitti, M., ... & Belladonna, M. L. (2021). Artocarpus tonkinensis extract inhibits LPS-triggered inflammation markers and suppresses RANKL-induced osteoclastogenesis in RAW264. 7. Frontiers in Pharmacology, 2417.

o Li, C. Y., Meng, Y. H., Ying, Z. M., Xu, N., Hao, D., Gao, M. Z., ... & Ying, X. X. (2016). Three novel alkaloids from Portulaca oleracea L. and their anti-inflammatory effects. Journal of Agricultural and Food chemistry, 64(29), 5837-5844.

o Yang, H., Xue, Y., Kuang, S., Zhang, M., Chen, J., Liu, L., ... & Deng, C. (2019). Involvement of Orai1 in tunicamycin-induced endothelial dysfunction. The Korean Journal of Physiology & Pharmacology, 23(2), 95-102.

Reviewer #2

This manuscript describes a series of studies looking at the anti-inflammatory effects of lipids extracted form Halocynthia aurantium. Overall the results are interesting, but revisions are needed to help improve the manuscript. These points are listed below.

1. The authors describe an extraction method in lines 86-98. This methodology needs to be more quantitative. How much material was used to perform the extract? What was the mass obtained in each fraction? This will allow one to know the lipid content in the extraction. While each fraction analyzed had the same mass of lipids, it would be more useful to have the number on a weight percent basis.

- It was corrected in Line 87-88 and 92-95.

- Total lipids produced a yield of 33.6 mg (w/w) or 0.75% of dry material (4.5 g). If 4.5 g of dried material is 100%, yield (33.6 mg) is 0.75% of dried material (4.5 g).

- The neutral lipids, glycolipids, and phospholipids were shown to have a high lipid yield of 8.63 (2.9 mg), 30.95 (10.4 mg), and 44.94 (15.1 mg) % of total lipid (33.6 mg), respectively.

- Thanks.

2. There needs to be a positive control in the bioassays.

- Normally the anti-inflammatory effects used LPS as positive control.

- Thanks. 

- References

o Li, M., Zhang, L., Cai, R. L., Gao, Y., & Qi, Y. (2012). Lipid‐soluble extracts from Salvia miltiorrhiza inhibit production of LPS‐induced inflammatory mediators via NF‐κB modulation in RAW264.7 cells and perform anti-inflammatory effects in vivo. Phytotherapy Research, 26(8), 1195-1204.

o Kim, K. N., Heo, S. J., Yoon, W. J., Kang, S. M., Ahn, G., Yi, T. H., & Jeon, Y. J. (2010). Fucoxanthin inhibits the inflammatory response by suppressing the activation of NF-κB and MAPKs in lipopolysaccharide-induced RAW 264.7 macrophages. European journal of pharmacology, 649(1-3), 369-375.

o Zhang, W., Yan, J., Wu, L., Yu, Y., Richard, D. Y., Zhang, Y., & Liang, X. (2019). In vitro immunomodulatory effects of human milk oligosaccharides on murine macrophage RAW264. 7 cells. Carbohydrate polymers, 207, 230-238.

3. Why was a 4.0% lipid concentration chosen for the results in Figure 2? No rationale for this choice is given and should be provided.

- Figure 2A showed cell proliferation in RAW264.7 cells (in the absence of LPS) that are toxic at high concentrations while cell proliferation in LPS-stimulated RAW264.7 cells showed non-toxicity.

- The purpose of our study is to test the anti-inflammatory effects. Therefore, we tested the three lipids with the addition of LPS. It showed the highest NO production at a concentration of 4%, especially phospholipids, which reduced NO production by 9.52 % while causing no toxic in the cells. 

- We changed the previous one to new figure 2A including the addition of LPS.

- All experiments with three lipids should use the same concentration.

- Following references showed the cell viability of a sample to be lower than control (≤ 80%), and that they can also be evaluated for NO production, gene expression, and protein expression.

- Thanks.

- References

o Qiao, J., Xu, L. H., He, J., Ouyang, D. Y., & He, X. H. (2013). Cucurbitacin E exhibits anti-inflammatory effect in RAW 264.7 cells via suppression of NF-κB nuclear translocation. Inflammation Research, 62(5), 461-469.

o Konishi, I., Hosokawa, M., Sashima, T., Maoka, T., & Miyashita, K. (2008). Suppressive effects of alloxanthin and diatoxanthin from Halocynthia roretzi on LPS-induced expression of pro-inflammatory genes in RAW264. 7 cells. Journal of Oleo Science, 57(3), 181-189.

o Orecchini, E., Mondanelli, G., Orabona, C., Volpi, C., Adorisio, S., Calvitti, M., ... & Belladonna, M. L. (2021). Artocarpus tonkinensis extract inhibits LPS-triggered inflammation markers and suppresses RANKL-induced osteoclastogenesis in RAW264. 7. Frontiers in Pharmacology, 2417.

o Li, C. Y., Meng, Y. H., Ying, Z. M., Xu, N., Hao, D., Gao, M. Z., ... & Ying, X. X. (2016). Three novel alkaloids from Portulaca oleracea L. and their anti-inflammatory effects. Journal of Agricultural and Food chemistry, 64(29), 5837-5844.

o Yang, H., Xue, Y., Kuang, S., Zhang, M., Chen, J., Liu, L., ... & Deng, C. (2019). Involvement of Orai1 in tunicamycin-induced endothelial dysfunction. The Korean Journal of Physiology & Pharmacology, 23(2), 95-102.

---

## [Decision Letter · Decision Letter 1]

20 Jun 2022

Anti-inflammatory effects of neutral lipids, glycolipids, phospholipids from Halocynthia aurantium tunic by suppressing the activation of NF-κB and MAPKs in LPS-stimulated RAW264.7 macrophages

PONE-D-22-03751R1

Dear Dr. Park,

We’re pleased to inform you that your manuscript has been judged scientifically suitable for publication and will be formally accepted for publication once it meets all outstanding technical requirements.

Kind regards,

C. Michael Greenlief, Ph.D.

Academic Editor

PLOS ONE

Additional Editor Comments (optional):

Reviewers' comments:

Reviewer's Responses to Questions

**Comments to the Author**

1. If the authors have adequately addressed your comments raised in a previous round of review and you feel that this manuscript is now acceptable for publication, you may indicate that here to bypass the “Comments to the Author” section, enter your conflict of interest statement in the “Confidential to Editor” section, and submit your "Accept" recommendation.

Reviewer #1: (No Response)

Reviewer #2: All comments have been addressed

2. Is the manuscript technically sound, and do the data support the conclusions?

Reviewer #1: Yes

Reviewer #2: Yes

3. Has the statistical analysis been performed appropriately and rigorously? 

Reviewer #1: Yes

Reviewer #2: Yes

4. Have the authors made all data underlying the findings in their manuscript fully available?

Reviewer #1: Yes

Reviewer #2: Yes

5. Is the manuscript presented in an intelligible fashion and written in standard English?

Reviewer #1: Yes

Reviewer #2: Yes

6. Review Comments to the Author

Reviewer #1: The revised manuscript answers all the questions well and the author also provided additional experimental data. I would like to recommend it for publication in PLOS ONE.

Reviewer #2: The authors have addressed all previous concerns. The authors have added some additional experimental work that helps to strengthen the manuscript.

7. PLOS authors have the option to publish the peer review history of their article (what does this mean?). If published, this will include your full peer review and any attached files.

Reviewer #1: **Yes: **Arjun H. Banskota

Reviewer #2: No

---

## [Editor Report · Acceptance letter]

5 Aug 2022

PONE-D-22-03751R1 

Anti-inflammatory effects of neutral lipids, glycolipids, phospholipids from *Halocynthia aurantium* tunic by suppressing the activation of NF-κB and MAPKs in LPS-stimulated RAW264.7 macrophages 

Dear Dr. Park:

I'm pleased to inform you that your manuscript has been deemed suitable for publication in PLOS ONE. Congratulations! Your manuscript is now with our production department. 

Kind regards, 

on behalf of

Dr. Charles Michael Greenlief 

Academic Editor

PLOS ONE